Early conch morphology of a gigantic Cretaceous ammonoid, Pachydesmoceras denisonianum (Desmoceratidae)

Wani Ryoji wani@ynu.ac.jp
Faculty of Environment and Information Sciences, Yokohama National University , Yokohama , Japan
Brygadyrenko Viktor
Electronic publication date: 2025 May 23
Publication date: 2025
Volume: 13
Electronic Location ID: e19488
Received 2025 Jan 27; Accepted 2025 Apr 27
Copyright: © 2025 Wani
Copyright year: 2025
Copyright holder: Wani
License: This is an open access article distributed under the terms of the Creative Commons Attribution License, which permits unrestricted use, distribution, reproduction and adaptation in any medium and for any purpose provided that it is properly attributed. For attribution, the original author(s), title, publication source (PeerJ) and either DOI or URL of the article must be cited.
License URL: https://creativecommons.org/licenses/by/4.0/

Keywords: Ammonoid, Conch morphology, Cretaceous

Funding: Grants-in-Aid for Scientific Research 21740369 and 22K03794 This work was supported by Grants-in-Aid for Scientific Research (no. 21740369 and 22K03794). The funders had no role in study design, data collection and analysis, decision to publish, or preparation of the manuscript.

==============================
Gigantic ammonoids, with conch diameters exceeding 1 m, remain one of the most enigmatic groups of extinct organisms. Their paleoecology has been the subject of ongoing debate, with some uncertainties arising from preservation biases, especially of an early conch. This study focuses on an exceptionally preserved early conch of the giant Cretaceous ammonoid Pachydesmoceras denisonianum from southern India. Conch morphology and the ontogenetic trajectories of constrictions and septal spacings were examined. The results indicate that constrictions were frequently present in the early conch; based on the shell layers observed in the cross-section, these constrictions likely resulted from periods of halted or slowed growth. The common occurrence of constrictions during early ontogeny suggests that Pachydesmoceras lifespan may have been longer than previously assumed. Additionally, the ontogenetic patterns of septal spacing might not reflect these growth halts or slowdowns in the examined species.

Introduction

Gigantic marine invertebrates evolved across several groups during and after the Cambrian explosion (Klug et al., 2015a). Ammonoids flourished from the Devonian to the end of the Cretaceous period and are a prime example of this trend. During this period, some ammonoid species grew to enormous sizes, with conch diameters exceeding 1 m, and these species are known worldwide (Klug et al., 2015a; Tajika, Nützel & Klug, 2018). One of the most notable examples is the Late Cretaceous Parapuzosia seppenradensis (Puzosiinae, Desmoceratidae, Perisphinctina; for higher taxonomy, see Besnosov & Mikhailova, 1991; Wright, 1996; and Yacobucci, 2015), which is currently recognized as the largest ammonoid species (Ifrim et al., 2021). Another prominent gigantic ammonoid is Pachydesmoceras (Puzosiinae, Desmoceratidae, Perisphinctina; Kin & Niedźwiedzki, 2012; Tajika, Nützel & Klug, 2018), which occurred from the upper Albian to upper Turonian in regions across Europe, Africa, Madagascar, India, Japan, and New Zealand (Wright, 1996). The type species of this genus is Pachydesmoceras denisonianum, with the lectotype (designated by Matsumoto, 1987) originating from the northeast of the village of Odiyam in the Ariyalur region of southern India.

The inner whorls of gigantic ammonoids are often preserved poorly, primarily because the earliest whorls tend to be dissolved during diagenesis (Maeda, 1987; Maeda & Seilacher, 1996; Maeda et al., 2010; Wani & Gupta, 2015). Ifrim et al. (2021) investigated the ontogeny of the largest ammonoid, Parapuzosia seppenradensis, including the early conch morphology (approximately 110 mm in conch diameter). In contrast, the internal conch morphology of Pachydesmoceras has rarely been documented, with P. pachydiscoide and P. kossmati being the exceptions (Matsumoto, 1988; Kennedy & Klinger, 2014).

During our fieldwork in the Ariyalur area of southern India (a collaborative fieldwork conducted with Dr. K. Ayyasami from the Geological Survey of India, authorized under permit no. 874/2002), an exceptionally preserved early conch of Pachydesmoceras denisonianum (the type species of Pachydesmoceras) was collected from a horizon nearly identical to that of its lectotype, close to the type locality. In this study, the outer and internal morphology of this specimen were examined to identify the early conch morphology of a gigantic ammonoid that has rarely been recognized in previous studies.

Materials

A single gigantic ammonoid specimen was discovered in the Karai Formation (Uttatur Group), located approximately 4 km southwest of Odiyam village (11°13′01′′N, 78°59′32′′E) in the Ariyalur area (Fig. 1; for detailed geological information, see Sundaram & Rao, 1986; Sundaram et al., 2001). The ammonoid, with a conch diameter of approximately 0.7 m, was unfortunately fragmented into several pieces, preventing the collection of the complete specimen. Only the nearly intact innermost part of the conch was retrieved in the field. The broken outermost whorl has a moderate whorl expansion ratio and a moderately wide umbilicus. The outermost whorl is ovoid in cross-section, with a rounded venter, convex flanks, rounded umbilical shoulder, and steep umbilical wall. Based on the conch morphology of the broken fragments, the large conch diameter, and the stratigraphic position, this specimen was identified as Pachydesmoceras denisonianum. The geological age of the specimen was considered to be late Albian–early Cenomanian, based on associated species from nearby localities, including Mariella bergeri, Puzosia compressa, and Mortoniceras spp.

Figure 1 Geological map around the specimen locality.

Modified from Sundaram et al. (2001). Solid star, specimen locality; white stars, village or town.

The outermost whorl is preserved without the shell as an internal mold, likely due to dissolution and/or peeling caused by the adjacent broken whorl. Consequently, the presence and prominence of ribs on the shell surface of the preserved outermost whorl could not be accurately assessed.

The specimen examined in this study is deposited in the Mikasa City Museum (MCM), Hokkaido, Japan, with the registered number (MCM-W2145). Additionally, for the comparison of conch morphology, the morphological data presented in Matsumoto (1988) were employed.

Methods

To observe the outer conch shape of the early conch, the specimen was first blackened using colloidal graphite and then whitened with ammonium chloride. In this study, conch diameter (D), conch diameter at 180° adapically from D (d), umbilical width (U), whorl height (H), whorl height at 180° adapically from H (h), whorl width (W), and aperture height (ah) were measured (Fig. 2). Then, simple ratios between two conch characters (U/D, U/H, W/D, and W/H), a whorl expansion rate (WER; (D/d)2), and an imprint zone rate (IZR; (H-ah)/H) were calculated. Conch terminology follows that of Klug et al. (2015b).

Figure 2 Measured conch morphology.

D, conch diameter; d, conch diameter at 180° adapically from D; U, umbilical width; H, whorl height; h, whorl height at 180° adapically from H; W, whorl width; ah, aperture height.

The examined specimen was subsequently polished along its median plane (plane of symmetry) using silicon carbide powder. Constrictions were observed on the median plane, and the spacing between successive constrictions was measured. These spacings are the rotational angle between two consecutive constrictions at the ventral positions. The center of rotation was defined as the center of the approximated logarithmic spiral. The relationship between the measured constriction spacings and conch diameter was examined throughout early ontogeny. Additionally, cross-sectional observations of the shell structure, particularly around the constrictions, were made.

The septal spacing between successive septa was also measured on the median plane. These spacings were defined as the rotational angle between two consecutive septa at the positions where the septum intersects with the siphuncle. Like modern nautilus, ammonoid growth involved continuous shell addition at the aperture to enlarge the body chamber, coupled with periodic secretion of septa behind the animal’s body (Hoffmann et al., 2015). This partitioned the older parts of the shell into gas-filled chambers used for buoyancy control, while the animal always lived at the growing front of the shell. Therefore, understanding the covariant development of septum and aperture is critical for reconstructing ammonoid growth and paleoecology (Bucher & Guex, 1990; Bucher et al., 1996; Bucher, 1997; De Baets et al., 2013). In this study, the relationship between the measured septal spacings and conch diameter was examined throughout early ontogeny.

The comparisons between constriction and septal spacings were statistically evaluated with t-tests, under the null hypothesis that the difference in slopes of spacing trajectories is zero.

Early conch morphology of pachydesmoceras denisonianum

Outer morphology of early conch

The early conch of the gigantic Pachydesmoceras denisonianum (109 mm in conch diameter) exhibits a discoidal shape (W/D = 0.38), is weakly compressed (W/H = 0.91), and has a moderate whorl expansion ratio (WER = 1.90) and a very wide and evolute umbilicus (U/H = 0.78) (Table 1; Fig. 3). The whorl is ovoid in cross-section, with a rounded venter, convex flanks, rounded umbilical shoulder, and steep umbilical wall.

Table 1 Morphological data of Pachydesmoceras denisonianum.

Specimen	D (mm)	d (mm)	U (mm)	H (mm)	h (mm)	W (mm)	ah (mm)	U/D	U/H	W/D	W/H	H/h	Whorl expansion rate (WER; (D/d)2)	Imprint zone rate (IZR; (H-ah)/H)	Reference	
Lectotype, GSI. 208	995		410	345	208	300		0.41	1.19	0.30	0.87	1.44			Stoliczka (1863–1866)	
GSJ. F3469	460		156	179	136	195		0.34	0.87	0.42	1.09	1.43			Matsumoto (1988)	
Yabe (1914), pl. 12	446		142	188		160		0.32	0.76	0.36	0.85				Yabe (1914)	
YKC. 610612	365		120	143	131	183		0.33	0.84	0.50	0.93	1.40			Matsumoto (1988)	
MNHN. 3750	166		52	67	45	65		0.31	0.78	0.39	0.97	1.43			Collignon (1961)	
MCM-W2145	109	79	35	45	31	41	32	0.32	0.78	0.38	0.91	1.45	1.90	0.29	This study	
Note:

Measurements except the examined specimen are from Matsumoto (1988). D, conch diameter; d, conch diameter at 180° adapically from D; U, umbilical width; H, whorl height; h, whorl height at 180° adapically from H; W, whorl width; ah, aperture height.

Figure 3 Photographs of the examined specimen, MCM-W2145.

(A) Left lateral view; (B, C) ventral views; (D) right lateral view. Stars are only shown in the left-side photograph, indicating the ventral positions of constrictions that can be recognized in the preserved outer whorl. Scale bar is 10 mm.

Constrictions are frequently observed, with 10 distinct constrictions identifiable on the preserved outermost whorl (Fig. 3A; Table S1). All constrictions are concave, prosiradiate, and project forward on the venter, exhibiting nearly uniform widths and depths from the umbilical seams to the venters. Due to the dissolution or peeling of the outer shell layer caused by the adjacent broken whorl, the presence and intensity of other ornamentations, such as ribs, remain uncertain.

Internal morphology of early conch

The earliest whorl is dissolved, and only approximately 1.5 whorls are preserved, with the smallest preserved conch diameter measuring 32 mm (Fig. 4). Thus, the earliest conch morphology, such as the ammonitella and initial chamber, cannot be observed or evaluated. The early conch follows a logarithmic spiral, with a moderately embracing imprint zone rate (IZR = 0.29; Table 1).

Figure 4 Cross section of the examined specimen.

(A) Photograph of the median section of the examined specimen; (B, C) enlarged photographs of constrictions and ribs shown in black squares and its schematic drawings. The exterior is toward the top of the photos. Black arrow, the preserved smallest conch; white arrows, the ventral position of constrictions; star, the smallest position of constriction that can be recognized in the preserved outer whorl; circle, the center of the approximated logarithmic spiral.

Twenty constrictions, including the 10 that are discernible on the outer morphology, are recognized at the median plane (Figs. 3, 4; Table S1). The conch diameters corresponding to the first and second constrictions are not measurable due to the dissolution of the earliest whorl. The smallest measurable conch diameter with a constriction is 33 mm (corresponding to the third constriction; Fig. 4; Table S1). Due to the dissolution of the earliest whorl, it remains unclear whether constrictions exist before the first observed constriction (Fig. 4). The rotational angles between successive constrictions of the 20 observed constrictions in the specimen ranged from 22° to 82° (average = 42.3°, standard deviation = 13.29°; Figs. 3–5; Table S1). Matsumoto (1988) reported that the body chamber length of the lectotype measures 240°. Assuming that this body chamber length remained consistent throughout the ontogeny of the examined specimen, the observed average constriction spacing of 42.3° corresponds to approximately one-sixth of the total body chamber length.

Figure 5 Ontogenetic trajectories of constriction and septal spacing.

(A) Graph of constriction spacing through early ontogeny; (B) graph of septal spacing through early ontogeny; (C) comparison between constriction and septal spacings. The apertural position at each septum was estimated under the assumption that the body chamber length remained 240° throughout ontogeny, consistent with the measurements of the lectotype (likely an adult macroconch; Matsumoto, 1988).

In the cross sections, certain ridges are discernible on the shell surface, which are associated with constrictions. In the present study, these features are referred to as ribs (Fig. 4). Two well-preserved constrictions reveal that ribs are positioned adapically to the constrictions in the cross sections (Fig. 4). Additionally, the outer shell layer at the constrictions is distinctly oblique to the shell surface. This feature is especially noticeable in the adoral parts of the constrictions (Figs. 4B, 4C). In contrast, the inner shell layer remains continuous, even across the constrictions.

The preserved first septum was located at a conch diameter of 38 mm (phragmocone diameter without the body chamber), and 25 septa were recognized in this specimen (Fig. 4; Table S1). The rotational angles between successive septa ranged from 18° to 26° (average = 21.9°, standard deviation = 1.93°; Figs. 4, 5; Table S1).

Discussion

Intraspecific and interspecific comparison of early conch morphology

Based on the measurements of Pachydesmoceras denisonianum, including the lectotype and specimen examined in this study (Table 1), intraspecific comparisons were made regarding the conch morphology at different conch diameters (Fig. 6). The graph between umbilicus width (U) and conch diameter (D) indicates that the umbilicus width/conch diameter (U/D) ratio tended to increase with growth, reflecting an enlarging umbilicus relative to the conch diameter (Fig. 6D). In contrast, the graphs of morphological ratios for whorl width (whorl width/whorl height ratio (W/H) and whorl width/conch diameter ratio (W/D); Figs. 6E, 6F) show greater variation in larger specimens, although the sample size is limited. These trends suggest that whorl width (W) exhibited more variation than other morphological parameters. This variability in whorl width may be attributed to the inflation of body chambers in Pachydesmoceras as it matured (Matsumoto, 1988), or to diagenetic deformation in larger specimens (Maeda et al., 2010; Klug et al., 2015a).

Figure 6 Measurements of Pachydesmoceras denisonianum.

(A) Conch diameter/umbilicus width relationship; (B) conch diameter/whorl height relationship; (C) conch diameter/whorl width relationship; (D) umbilicus width/conch diameter ratio (U/D); (E) whorl width/whorl height ratio (W/H); (F) whorl width/conch diameter ratio (W/D). Solid circle, measurements of lectotype; multiple mark, the examined specimen.

A relatively smaller conch of P. denisonianum (166 mm in diameter; Table 1) was reported by Collignon (1961, pl. 8), collected from the Cenomanian of Madagascar. According to Matsumoto (1988), this specimen displays crowded ribs of unequal length, with some of the longer ribs accompanied by indistinct constrictions. In the specimen examined in this study, ribs are not visible on the shell surface, likely due to the preservation of the outer shell layer. Given the relationship of ribs accompanied by constrictions observed in Collignon’s specimen (Collignon, 1961), a similar relationship is likely present in the examined specimen.

At a later stage in P. denisonianum, Matsumoto (1988) noted that constrictions became less distinct and may only appear as shallow furrows along some of the longer ribs. Considering this observation alongside the early conch morphology identified in this study, it can be inferred that the frequency and prominence of constrictions would decrease with growth in P. denisonianum.

Early conch morphology (conch diameter < ~150 mm) has been recognized in P. pachydiscoide and P. kossmati (Matsumoto, 1988; Kennedy & Klinger, 2014). According to Matsumoto (1988) and Kennedy & Klinger (2014), early conchs of both species exhibit frequent constrictions associated with ribs, which later become less distinct. In contrast, the ribs become more prominent, thicker, and coarser in the later stages. In the mature stage, the ribs gradually weaken, resulting in a nearly smooth conch (Matsumoto, 1988). From the early to later stages, these ontogenetic trends in shell ornamentation are similar to those observed in P. denisonianum (early conch in this study and later conch in Matsumoto, 1988).

Implication for shell growth from constrictions and associated ribs

The observation of the cross section of the examined specimen revealed that ribs are located just adapically to the constrictions (Fig. 4). The nearly identical width and depth of each constriction from umbilical seams to venter (Fig. 3) suggests that the ribs at these positions extend along the constrictions, resulting in long ribs, which are ribs extending from umbilical seams to venter (see also Matsumoto, 1988). Similarly, in the relatively smaller specimen from the Cenomanian of Madagascar (Collignon, 1961), some long ribs are accompanied by indistinct constrictions (Matsumoto, 1988). Matsumoto (1988) noted that long ribs begin to appear at least by the middle stage in Pachydesmoceras. In later stages, these long ribs are spaced at gradually broadening intervals, with shorter ribs intercalated. However, this study implies that long ribs appear as early as the initial stages, at least when the conch diameter reaches 33 mm.

An examination of the well-preserved constrictions revealed that the outer shell layer at these constrictions is distinctly oblique to the shell surface, which is especially noticeable in the adoral parts of the constrictions (Fig. 4). The discontinuous shell layers across the constrictions and ribs suggest a transition in growth phases at these points, potentially indicating a growth halt or slowdown. Similar distinct shell layers have been reported by Bucher et al. (1996), who studied the shell layers associated with conch ornamentation called megastria. This thick, continuous line extends around the flanks and venter of an ammonoid conch. Their study observed discontinuous outer shell layers and continuous inner shell layers at the megastria, concluding that such discontinuities represented growth halts. Based on these observations, constrictions have also been hypothesized to be associated with growth halts or slowdowns (Arkell, Kummel & Wright, 1957; Kulicki, 1974; Kennedy & Cobban, 1976; Obata et al., 1978; Westermann, 1990; Bucher et al., 1996; Bucher, 1997; De Baets et al., 2013; Klug et al., 2015b; Urdy, 2015). The findings in this study align with this hypothesis, although shell abnormalities would be another interpretation for constrictions (Landman & Waage, 1986). In this respect, the constrictions in this study resemble megastriae in Bucher et al. (1996), which extend continuously around the flanks and venter of the shell and are interpreted as markers of intrinsic growth pauses. The repeated occurrence of constrictions in the early conch (at least 20 constrictions up to a conch diameter of 109 mm) suggests that the examined specimen experienced several growth halts or slowdowns during early ontogeny. As constrictions become less frequent in the later stages of Pachydesmoceras, growth halts, or slowdowns may decrease with growth. However, the frequency of constrictions could vary between specimens or species. Since only a single specimen was analyzed in this study, further investigation with additional specimens is necessary to determine whether the frequency of these growth halts or slowdowns are species-dependent, genus-dependent, or environmentally influenced (e.g., seawater temperature, chemical composition of seawater, nutritional condition, and oxygen condition).

Constrictions are commonly observed on the conchs of Pachydesmoceras and other ammonoids, in contrast to Cretaceous nautiloids from the same region, which do not exhibit constrictions except a nepionic constriction formed at hatching (Blanford, 1862; Stoliczka, 1863–1866; Wani & Ayyasami, 2009; Wani, Kurihara & Ayyasami, 2011). This difference may be attributed to growth halts or slowdowns in ammonoids, as opposed to nautiloids, which lack such features. On the other hand, growth rates in modern nautiluses are variable and the growth rate decelerates during ontogeny (Landman & Cochran, 1986; Landman, Arnold & Mutvei, 1989; Cochran & Landman, 1993). However, we do not see a huge expression of this in the shells (like apertural constrictions). This may imply an alternate possibility that apertural constrictions could be produced by another pathway.

Given the potential repeated halts or slowdowns in shell growth during the early ontogeny and the large conch diameter of P. denisonianum, it is likely that their lifespan was longer than previously assumed. However, accurately estimating their life duration remains challenging at present. Isotopic sclerochronology could provide valuable insights into this question; however, the state of shell preservation in the examined specimen is insufficient to permit such analysis.

Ontogenetic trajectory of septal spacing

The measured septal spacings of the early conch in Pachydesmoceras denisonianum reveal a nearly stable pattern with a slightly increasing trend (standard deviation = 1.93°) (Fig. 5B). The precise correlation between the timing of apertural and septal formation remains unclear. However, Matsumoto (1988) reported that the body chamber length of the lectotype is 240°. Based on this, the apertural position at each septum was estimated in the present study, assuming that the body chamber length remained constant at 240° throughout the ontogeny of the examined specimen (Fig. 5C; Table S1). Figure 5C illustrates that, despite a decrease in constriction spacings (conch diameter ~70 mm), the septal spacing continues to exhibit a stable increasing trend. The overall increasing trends observed in both septal and constriction spacings (>60 mm in conch diameter) appear broadly similar (Fig. 5C), with the exception of a marked decrease in constriction spacing at a conch diameter of approximately 70 mm (t-test, t = −1.82, df = 16, p = 0.09). Given the average spacings of constrictions (42.3°) and septa (21.9°), it is likely that approximately two septa formed during each growth interval between successive constrictions. However, this proportional relationship does not hold at the ontogenetic stage corresponding to the ~70 mm conch diameter, where a decrease in constriction spacing occurred without a corresponding fluctuation in septal spacing (Fig. 5C). When the full dataset is considered (including the anomalous decrease at ~70 mm), the similarity between the overall trends in septal and constriction spacings is statistically rejected (t-test, t = −2.63, df = 18, p = 0.02). This suggests that, at this specific ontogenetic stage, more closely spaced constrictions are developed despite relatively consistent septal spacing. These findings indicate that constrictions and septal spacings are not always directly linked in Pachydesmoceras (Fig. 5C). This lack of connection implies that growth halts or slowdowns were not recorded in the ontogenetic trajectories of septal spacings, at least in the visible parts of Pachydesmoceras. Comparable analyses of septal spacing in relation to conch morphology and ornamentation have been previously reported (Bucher & Guex, 1990, Bucher et al., 1996, Bucher, 1997; De Baets et al., 2013), suggesting that growth pauses appear to be reflected in septal spacing.

The ontogenetic trajectories of Pachydesmoceras denisonianum were compared with those of other ammonoids from the subfamilies Puzosiinae (e.g., Puzosia sp., from the Turonian of the Ariyalur area, southern India) and Desmoceratinae (e.g., Desmoceras latidorsatum var. media, from the Albian of the Mahajanga area, Madagascar), both within the family Desmoceratidae (Takai et al., 2022; Nishino et al., 2024; Fig. 7). Although the comparable conch diameters among the examined three taxa are limited, the observed ranges of septal spacings in Pachydesmoceras fall within those of Desmoceras (especially those with phragmocone diameters >1 mm). The slope of the slightly increasing trend in Pachydesmoceras is almost parallel to that of Puzosia (t-test, t = 0.41, df = 80, p = 0.69). These suggest that the septal spacings of Pachydesmoceras share characteristics with both taxa. These trends in the Puzosiinae and Desmoceratinae are similar to those observed in other Cretaceous Perisphinctina but differ from those seen in Phylloceratina and Lytoceratina from the Cretaceous period (Arai & Wani, 2012; Iwasaki, Iwasaki & Wani, 2020; Kawakami, Uchida & Wani, 2022; Takai et al., 2022; Kawakami & Wani, 2023; Nishino et al., 2024).

Figure 7 Comparison of ontogenetic trajectories of septal spacings in the subfamilies Puzosiinae and Desmoceratinae.

Puzosia and Desmoceras data are from Nishino et al. (2024) and Takai et al. (2022), respectively.

Growth halts in ammonoids, which may be accompanied by constrictions, likely result from the covariance between isometric or allometric growth of the aperture and ornamentation (Bucher, 1997) and/or the balance between conch and soft part growth. The results of this work indicate that when the growth balance between the conch and soft parts in Pachydesmoceras was disrupted, this imbalance may have been compensated not by changes in septal spacing, but by modifications in the apertural shape, leading to the formation of constrictions. Furthermore, this imbalance may have occurred with relative consistency, at least during a specific early ontogenetic stage of the examined gigantic ammonoid species. Whether this phenomenon was common across ammonoids remains uncertain, and therefore, the relationship between the ontogenetic trajectories of constrictions and septa in various taxa across different geological ages warrants further investigation in future studies. This would provide a deeper understanding of how growth patterns and environmental factors influenced the morphological evolution of ammonoids over time.

Supplemental Information

Supplemental Information 1 Raw data of Pachydesmoceras denisonianum conch morphology.

I sincerely thank the Director General of the Geological Survey of India (Kolkata), K. Ayyasami, and S. Anantharaman for their kind help during fieldwork and fossil sampling in the Ariyalur area, southern India. Without their assistance, the fieldwork would not have been possible. I also thank A. Tajika and D. Aiba for their support during literature reviews and E. Yazykova, K. De Baets, D. Petermann, C. Ifrim, and an anonymous reviewer as well as the editor (V. Brygadyrenko) for their valuable and thoughtful comments on an earlier draft of the manuscript.

Additional Information and Declarations

Competing Interests

The authors declare that they have no competing interests.

Author Contributions

Ryoji Wani conceived and designed the experiments, performed the experiments, analyzed the data, prepared figures and/or tables, authored or reviewed drafts of the article, and approved the final draft.

Data Availability

The following information was supplied regarding data availability:

The raw data is available in the Supplemental File.

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
