# Peer review of "Early conch morphology of a gigantic Cretaceous ammonoid, Pachydesmoceras denisonianum (Desmoceratidae)"

_PeerJ, doi:10.7717/peerj.19488_

## Round 0.1 · original submission · Major Revisions

Dear Dr. Wani, I ask you to carefully study the reviewers' fundamental comments and improve the manuscript. I hope that the new version of this article will be approved by the reviewers.

·

Basic reporting

I guess this is an interesting study and results. However, excluding the importance of environmental items is a bit curious. I think you should open the discussion without categorical suggestions. I am not an English speaker but I made some corrections. I also added the paper which should be included. If you need the PDF do not hesitate to ask me directly. I also corrected the name of one author in the text and the reference list. I've got the PDF of this one too: Besnosov (not Bessenova) & Mikhailova, 1991.

Experimental design

All my comments you will find in the text

Validity of the findings

I think this study should be published after some modifications

Additional comments

no comment

·

Basic reporting

The study reports new findings of the early conch morphology of Pachydesmoceras. These are based on the rare discovery of a suitable preserved specimen for this purpose. I feel the literature is mostly referenced appropriately and sufficient background/context is provided. The only exception are information about possible diagenetic transformations of larger specimens (e.g., Klug et al. 2015) as well as the importance of comparing septal and constriction spacing at a distance of one body chamber length (e.g., Bucher 1996; De Baets et al. 2013 and references therein).

Klug, C., De Baets, K., Kröger, B., Bell, M. A., Korn, D., & Payne, J. L. (2015). Normal giants? Temporal and latitudinal shifts of Palaeozoic marine invertebrate gigantism and global change. Lethaia, 48(2), 267-288.

Experimental design

The author seemingly compares the spacing of constrictions and septa as if they were in the same position (at least it is not clearly discussed a body chamber length offset was used). According to the literature (e.g., Bucher et al. 1996; De Baets et al. 2013), it is considered more appropriate to compare the constriction at the aperture aperture with the septal spacing at a distance of one body chamber length apart. If the body chamber length is unknown, there are known correlations between body chamber length and conch shape which could be used as an approximation. If such a comparison is not or only partially possible, this limitation should at least be discussed. It will affect its explanatory power.
As your specimen is incomplete - correlation or lack thereof could be due to missing inner whorl as well as outer whorl(s).

De Baets, K., Klug, C., Korn, D., Bartels, C., & Poschmann, M. (2013). Emsian Ammonoidea and the age of the Hunsrück Slate (Rhenish Mountains, Western Germany). Palaeontographica A, 299(1-6), 1-113.

Validity of the findings

I feel findings are mostly carefully discussed and appropriate. I only feel that the interpretations would be (even) more robust when more appropriate comparisons (e.g., body chamber length between septa and constriction spacing analyses) or quantitative comparisons (e.g., when comparing the trajectories of Pachydesmoceras with those of others - ordination or cluster analyses may be useful in these endeavor) would be made. Also visual comparisons (e.g., Fig. 5) would benefit to focus on those part of the trajectories which is preserved in your specimen (e.g., maybe add a zoom on the graph above 10 mm)

Additional comments

Line 44: Where special permissions for field work in India needed. Please specify.
Line 65: Should be "is deposited" as it concerns only a single specimen. More importantly, please specify the collection/repository number of the specimen.
Line 111: Please add "first observed/observable constriction" to avoid confusion with the initial constriction.
Line 135: Please provide reference(s) supporting the statement - diagenetic deformation of larger specimens
Line 151: "less distinct" may be more appropriate then "more indistinct"

---

## Round 0.2 · Minor Revisions

Dear Dr. Wani, I ask you to carefully correct the reviewers' comments and hope that the new version of this article can be approved by them for publication.

·

Basic reporting

In general, the manuscript has been revised correctly. However, I still guess that the locality map is necessary and the explanation figure of conch measurements would also be useful. This is not comfortable for readers looking for some additional references just to find the geographical location or scheme of measurements. The village is not a well-known place, and such a scheme is not common. I am sure the paper is loosing without these two illustrations.

Experimental design

No comments

Validity of the findings

No comments

Additional comments

No comments

·

Basic reporting

The revised manuscript has significantly been improved and i look forward to seeing it published. There are still some supporting references missing for particular statement.

Experimental design

Explanations on the importance and why comparison of the septal and constriction spacing at body chamber angles and their potential implications should be discussed more extensively in methods and supporting reference of previous work should be cited in this context. It is quite easy to do (see annotated pdf).

Validity of the findings

The underlying data is appropriate, provided and appropriately visualised. I still feel the interpretations would be more robust if a statistical comparison would be made between the average septal spacing at similar diameters of Pachydesmoceras, Desmoceras and Puzosia. I understand the sample of Puzosia is limited but i would argue it makes it more rather than less important to do statistical tests. Either way - i leave it up to the author and the editor to decide on this but if no additional statistics are made - at minimum the limited overlapping preserved part with constrictions / septal spacing and potential implications on interpretations should at be more carefully discussed and highlighted (see suggestions in annotated pdf).

Additional comments

Suggestions concerning previous comments and additional points can be found as comments in the annotated pdf.

·

Basic reporting

This study involves the ontogeny and morphogenesis of a gigantic ammonoid. Features in its early ontogeny (apertural constrictions) are used to suggest periodic growth halts could have contributed to a longer lifespan in these taxa. I feel the discussion could be strengthened by elaborating on several open-ended questions concerning morphogenesis as well as a clear statement of future goals and approaches that can potentially address these questions.

I have included line-by-line comments and other notes below:

Line 24: Why does the reference “Klug et al., 2015b” appear before 2015a? Please reverse the lettering in the text and reference list.

Line 115: It would be helpful to also report this average angle between constrictions in terms of angular body chamber lengths. I.e., 42.3 / 240 = ~17.6% of the angular body chamber length. Consider adding a short note on this in the discussion – that growth in between these constrictions represents roughly one fifth of the angular body chamber length.

Line 124: It also might be useful to mention this average septal spacing angle in the discussion as a proportion of the average constriction spacing. I.e., Roughly two septa were formed at the growth interval in between the constrictions.

Regarding the two points above: In the discussion, consider comparing these constriction spacings to other ammonoid taxa. For example the Wocklumeriidae commonly have constrictions with much larger spacings (120 degrees, and very consistently spaced). But it seems the investigated species is on par with other desmoceratoids (e.g., several kossmaticeratids).

Line 192: Regarding nutritional conditions and metabolism – Nautilus can often be a poor analog for ammonoids. However, some comparisons can still be drawn regarding conch morphogenesis. It turns out Nautilus can have extremely variable periods of growth with septal formation taking anywhere from two weeks to several months (see Landman and Cochran, 2010, Growth and Longevity of Nautilus, Chapter 28 of “Nautilus: the biology and paleobiology of a living fossil”). It may be worth noting that we don’t see any extreme expressions of these hiatuses in Nautilus.

Line 202: Would isotope sclerochronology be useful to answer this question? I realize your material might not have the best preservation, but could this hypothesis be tested with well-preserved aragonitic shell material in another study?

Line 214 and Fig. 3C: I’m curious about this comparison between septal spacing and constriction spacing. I would expect the drop in constriction spacing to correspond to a potential drop in septal spacing, specifically for the septa 240 degrees behind the constriction (i.e., septa at comparatively lower whorl heights). That is, if there was a shortened growth interval in between two constrictions, we would expect that to crowd septa, or else the body chamber ratio would be reduced, causing problems with hydrostatics (like buoyancy and mass distribution). Alternatively, there could be corresponding changes in shell proportions, like whorl width and height, that reconciled these differences in body chamber volume relative to the whole animal volume. Even if your current dataset cannot confirm this trend, it may be worth adding a short note on these relationships to the discussion. It could also be an interesting area for future work on ammonoid species that might have more accessible material.

Experimental design

I would like to see more elaboration on Fig. 3C in the text and caption. At first glance, it was hard to tell that the conch diameter corresponding to the septal spacings do not correspond to the conch diameter at those septa, but rather the conch diameter at the constriction, with septa measured one body chamber length back (i.e., 240 degrees). This could be made clearer for future readers.

Validity of the findings

I've included some info related to the ambiguity of these constrictions:

Similar features to these constrictions appear in some scaphitid ammonoids as pathologies or abnormalities. These features are considered stretch phenomena, interpreted as rapid growth areas rather than hiatuses. While apertural constrictions in other taxa represent consistent morphological characteristics, rather than abnormalities, it might be worth mentioning this for completeness because it is an opposite interpretation of growth rates. See Landman and Waage (1986) Shell abnormalities in scaphitid ammonites, Lethaia.

In the title and abstract, emphasis was placed on the size of this ammonoid, but this was not revisited in the discussion. Gigantic ammonoids are somewhat uncommon compared to all other species. Additionally, they pose an interesting scaling relationship associated with the rates of cameral liquid transport during chamber emptying. The surface area of the siphuncle grows by L^2, but the cameral volumes grow by L^3, suggesting it may have taken longer for very larger chambers to be emptied (especially since siphuncle proportions do not really compensate for this with allometry; see Chamberlain and Moore, 1982, Paleobiology, 8(4): 408-425). We would expect growth halts related to this scaling limitation at larger sizes, however, these constrictions commonly attenuate throughout ontogeny in desmoceratoids. Furthermore, Nautilus growth halts are not really expressed in its conch morphology.

Additional comments

Line 337 and elsewhere: Please correct “Urgy” to “Urdy”.

Reviewer 4 ·

Basic reporting

Overall, the manuscript is clearly written; however, I believe certain sections would benefit from additional clarification and refinement to strengthen the rigor and scientific soundness of the findings.

Experimental design

This study relies on descriptive methods applied to a single specimen of a large ammonoid species. Although the approaches used by the author are relatively straightforward, they are fundamental for accurately characterizing ammonoid features.

Validity of the findings

Although I do not fully concur with some of the author’s interpretations and conclusions, I recognize that certain aspects are difficult to verify with the current data. Still, describing uncommon specimens such as the one examined here is valuable for advancing our understanding.

Additional comments

This manuscript examines the early ontogeny of a large ammonoid identified as Pachydesmoceras denisonianum. The author interprets that the constrictions visible both on the outer shell and in the cross-section represent growth halts and thus, indicate an extended lifespan in this species. To my knowledge, the early ontogeny of large ammonoids is generally only poorly preserved and thus, providing new data concerning early morphology is important. Overall, I recommend this paper for publication. However, I also believe that certain sections could be clarified and refined further. I have several comments, suggestions, and questions. Please see my detailed remarks below.

Line 2 Title: I suggest including the full species name and its author citation, as this paper specifically focuses on a single specimen of the species rather than the entire genus.
Line 19-20 “Additionally, the ontogenetic patterns of septal spacing do not appear to reflect these growth halts or slowdowns.” I am not completely sure about this. A quantitative/statistical test might help to clarify this further. Please see my comment below.
Line 58-59: The author identified this species as Pachydesmoceras denisonianum based on the conch morphology of the larger part of the shell that is not examined in this study. I suggest expanding the discussion of how this species was determined by providing more specific description about this specimen, diagnostic characters of this species, and relevant comparisons. Providing photographs of the large shell fragments would be helpful in supporting this identification.
Line 36 “outer shell layer”: Does it mean the outer prismatic layer of the shell or the specimen is preserved without the shell as an internal mold? Please clarify.
Line 90 “B”, “D”, “H”: Please introduce these abbreviations in the main text before using them. Although they are explained in the caption of Table 1, it is essential for clarity and readability that they be defined at their first mention in the text as well.
Lines 113-114: I wonder if the constrictions on the shell surface and those observed in the cross-section are directly comparable. It is well documented that the primary constriction seen on the shell surface corresponds to the transition from the embryonic to the post-embryonic shell in the cross-section. However, have any studies examined the correlation between these features in later ontogenetic stages? If so, please consider including references to support this point.
Line 124: I think Fig. 4 should be added here and to Line 115.
Lines 140-141 “Collignon (1961, pl. 8)”: Looking at the Collignon’s figure illustrating this specimen, I do not see any constrictions.
Lines 143-144: What is “the relationship between constrictions and ribs”? I suggest describing it more specifically.
Line 151: What does “Early conch” indicate here? At about what conch diameter? Is it comparable to the size of the studied specimen?
Line 156-157 “From the early to…in P. denisonianum”: Has the manuscript specifically discussed any ontogenetic changes in rib patterns? While Fig. 4 illustrates the ontogenetic variations of certain conch parameters, it does not include data on ribs. As a result, the statement regarding rib changes seems unsupported by the current results and data.
Line 157-159 ”Therefore, observing…related genera.”: It appears that this statement is not supported by any data or prior discussion in the manuscript. The previous sentence only notes that the ontogenetic trend is similar among P. pachydiscoide, P. kossmati, and P. denisonianum, without offering additional information regarding taxonomy and systematics. I recommend providing further details or evidence to support this claim and clarify its taxonomic and systematic implications.
Line 162 “ribs”: In Fig. 2, the structure labeled as a “rib” appears to be approximately 0.3 mm in size. Given such a small scale, is it truly appropriate to classify this feature as a rib, or might it be more comparable to a growth line, lira, or megastria based on its degree of “bump”? I would appreciate further discussion or clarification on this point.
Line 164: Why is the “long ribs” placed in the quotation marks? Does it have a special meaning? Additionally, I do not see the connection between “ the nearly identical width and depth of each constriction” and the “the ribs at these positions extend along the constrictions). An expanded explanation of how these observations relate to one another would be helpful.
Line 166: As I mentioned above, I do not see constrictions in the figure of the Collignon paper. There seems to be also no description about constrictions in the description part of the paper (p. 39).
Line 169 ”based on the current observation”: It appears that the presence of long ribs at 33 mm was inferred rather than directly observed by the author. I recommend revising the text to clarify that this is an assumption, not an observed fact.
Line 174-176: What is the difference between the “ribs” in this study and megastriae in Bucher et al. (1996)? Why are the “ribs” not considered megastriae? I would appreciate a discussion on this.
Line 183: Not “Urgy” but “Urdy”.
Line 199 “Given the repeated halts or slowdowns”: I suggest adding “potential” here to indicate that this is not certain.
Line 200-201: Could the author clarify what the preciously assumed lifespan of this species was and how it might be extended by the potential growth interruptions? Any additional insights on these estimates would be appreciated.
Line 211-212: In my view, the pattern of constriction and septal spacing over the overlapping conch diameter range appears comparable. If the apparent outlier at approximately 70 mm is excluded, both show a similar increasing trend. It might be valuable to conduct a statistical comparison to confirm this observation.
Line 237: Why does the formation of constrictions appear relatively consistent at least within a given ontogenetic stage? Does this suggest that soft tissue disruptions might occur on a regular, periodic basis? It would be helpful if the manuscript expanded on these points in more detail.
Fig. 1: Including photographs of larger fragments that were used to identify the species would be helpful.
Fig. 4: Since these data are primarily drawn from previous studies, please label and indicate each dataset accordingly in both the figure and its caption. The datapoint from this study should also be distinctly identified using e.g., a different symbol or marker to avoid confusion and clearly separate new data from those in the literature.

·

Basic reporting

As usual from this author this is a concise, well concepted study.

The suggestions by the reviewers were all addressed. Unfortunately they are not given in original, so sometimes it is impossible for me to find out why the author did not follow it.

A few minor points:
when found an with D around 0.7 m, was it with or without body chamber? With would not be giant. Is there a field photo?
What are large sepcis and diameters in the Aptian-Albian at all? I lack comparison here.

One small point:
I wondered about this sentene in the introduction:
(Puzosiinae, Desmoceratidae,
29 Perisphinctina; for higher taxonomy, see Besnosov & Mikhailova, 1991, and Yacobucci, 2015),
Yaccobucci, 2015 is certainly not the primary source, her study is based onthe Treatise on invertebrate palaeontology Cretaceous ammonoids volume which should be cited as reference here. Why do the authors follow Besnosov & Mikhailova in higher taxonomy and not others? in addition, this is of no importance for the interpretations, and there is no urgent need to mention this at all.

The is one more point from my side:
I would like to see a localtiy map, preferably on a geological map, and, if possible, a section; if not in the main part then in the supplement. Such additional field info may later be of importance for metastudies. To my eyes it is not sufficient to refer to another reference.

Fig 3c repeats 3a, making the latter obsolete. Why not linear scale?

Fig 5, could be the trajectories separated? Where does that of Pachydesmoceras begin?

Experimental design

The research question is addressed carefully and supperted by primary data - great! The conclusions, however, remain rather open to further studies, although this approach is much better than overinterpretation. In this resepct this study, however, loses a bit in importance. Maybe a research question should be developed and then answered in the conclusions.

Validity of the findings

This is a very detailed description of morphology of only one species and specimen at a high academic level, but without deeper-going conclusions. Whether this is within the scope of the journal is a decision of the editor.

---

## Round 0.3 · Minor Revisions

Dear Dr. Wani, I ask you to make minor corrections to the manuscript.

·

Basic reporting

Sufficient field background/context is now provided. Raw data shared.

Experimental design

Research question is well defined. It is also clearly stated how this new discovery fill the identified knowledge gap. Methods are now also described in detail which allows replication.

Validity of the findings

Conclusions are clearly stated and supported by data and analyses.

Additional comments

Thank you for addressing all of the (remaining) points me and other reviewers have raised. I feel the revised manuscript is ready for publication.

·

Basic reporting

I am happy with the revised manuscript. See "additional comments" section

Experimental design

I am happy with the revised manuscript. See "additional comments" section

Validity of the findings

I am happy with the revised manuscript. See "additional comments" section

Additional comments

I am happy with the changes in the current version.

However, I still urge the author to have a brief note about modern Nautilus. Growth rates in Nautilus are variable. It often can halt growth for very long periods, however, we do not see a huge expression of this in the shell (like apertural constrictions). I do not think this point undermines the study because there are considerable differences between extant nautilids and the ammonoids. However, I think it would be useful to add for completeness because it mentions the alternate possibility that apertural constrictions could be produced by another pathway.

Reviewer 4 ·

Basic reporting

I appreciate the author’s efforts to revise the manuscript. It has substantially improved. I have just a few points that I believe are important. I think the manuscript will be ready for publication once these are addressed.

Experimental design

I have no additional remarks here.

Validity of the findings

I have no additional remarks here.

Additional comments

Lines 60-62: As noted in my previous review, I remain concerned about reproducibility, since the specimen and any photographic documentation are unavailable. The authors should provide a more detailed description of the specimen and specify the diagnostic characters used to assign it to this species (how was the specimen assigned to this species?). Given that the species name appears in the title, this information is particularly crucial.
Lines 247-267: Please consider adding figure numbers to relevant sentences to enhance readability.
Line 250: I appreciate the addition of statistical tests. However, I strongly suggest expanding the Methods section to describe exactly what each of the two t‑tests examined and the hypotheses they address. Additionally, please report exact p‑values rather than using the vague “p > 0.05.”

---

## Round 0.4 · accepted · Accept

Dear Dr. Wani, I congratulate you on the acceptance of this article for publication and hope that you will continue to research this topic and send more of your articles to our journal.

Reviewer 4 ·

Basic reporting

I appreciate that the author has addressed all previous comments and suggestions raised by the reviewers. I believe that the manuscript is now ready for publication.

Experimental design

I have no additional remarks here.

Validity of the findings

I have no additional remarks here.

Additional comments

I have no additional remarks here.